# Differentiation between Normal Cognition and Subjective Cognitive Decline in Older Adults Using Discrepancy Scores Derived from Neuropsychological Tests

**DOI:** 10.3390/geriatrics9030083

**Published:** 2024-06-19

**Authors:** Ramón López-Higes, Susana Rubio-Valdehita, Sara M. Fernandes, Pedro F. S. Rodrigues

**Affiliations:** 1Departamento de Psicología Experimental, Complutense University of Madrid (UCM), 28223 Madrid, Spain; rlopezsa@ucm.es; 2Departamento de Psicología Social, del Trabajo y Diferencial, Complutense University of Madrid (UCM), 28223 Madrid, Spain; srubiova@ucm.es; 3CINTESIS@RISE, CINTESIS.UPT, Portucalense University, 4200-072 Porto, Portugal; sarab@upt.pt

**Keywords:** subjective cognitive decline, discrepancy scores, sentence comprehension, naming, memory, executive functions

## Abstract

Several studies have reported subtle differences in cognition between individuals with subjective cognitive decline (SCD) compared to those with normal cognition. This study aimed to (i) identify these differences using discrepancy scores (e.g., categorial–phonemic verbal fluency performance) derived from neuropsychological tests in three cognitive domains (memory: Wechsler’s Word List and Digits; executive functions: Stroop and verbal fluency; and language: BNT and ECCO_Senior) and (ii) determine which discrepancy scores are significant for classification. Seventy-five older adults were included: 32 who were labeled SCD+ (age 71.50 ± 5.29), meeting Jessen et al.’s criteria, and 43 in the normal cognition group (SCD−; age 69.81 ± 4.62). Both groups completed a protocol including screening and the specified neuropsychological tests. No differences were found between the groups in their age, education, episodic memory, global cognitive state, or mood. Significant differences between the groups were observed regarding the discrepancy scores derived from BNT (naming) and ECCO_Senior (sentence comprehension). These scores accurately classified participants (71.6%), with ECCO_Senior having a primary role. ROC curves indicated a poor-to-fair model quality or diagnostic accuracy (AUC__BNT_ = 0.690; AUC__ECCO_ = 0.722). In conclusion, discrepancy scores in the language domain are important for distinguishing between individuals with SCD and normal cognition, complementing previous findings in this domain. However, given their relatively poor diagnostic accuracy, they should be used with caution as part of a more detailed neuro-psychological assessment.

## 1. Introduction

Subjective cognitive decline (SCD) is associated with a three-to-six-fold increased risk of clinical progression to mild cognitive impairment (MCI) in cognitively healthy individuals [1,2]. However, SCD can vary in stability over time; some individuals may exhibit fluctuations in cognitive decline progression [3]. Additionally, there is no accepted gold standard definition of SCD. Jessen et al.’s inclusion criteria (2014) [4] for individuals with SCD set that they must have a self-experienced persistent decline in their cognitive capacity in comparison to a previously normal status and be aware of these changes. Subjective concerns expressed about cognitive decline, the onset within the last five years, feelings of a worse performance compared to peers, and related anxiety or depression are supportive evidence. These individuals show a cognitive performance within the normal range in standardized cognitive tests and the absence of major psychiatric, neurological, or medical disorders affecting cognition. Jessen et al.’s proposal captures subjective experiences, integrates temporal and emotional factors, and includes additional research criteria like biomarker presence, genetic risk factors, as well as the influence of lifestyle factors, thus providing a multi-dimensional approach. However, the complexity and the need for a detailed assessment can make it difficult to apply this in routine clinical settings. Additionally, it has been argued that a biomarker assessment is not always feasible in all clinical contexts. Reliance on patient self-reporting may introduce biases.

Petersen et al. (2014) [5] suggested that SCD should be considered a preclinical stage of Mild Cognitive Impairment (MCI) and Alzheimer’s Disease (AD), emphasizing the need for a longitudinal follow-up to track conversion from SCD to MCI or AD. This proposal highlights the potential significance of SCD as an early marker for neurodegenerative diseases. However, it focuses more on the progression to AD, possibly overlooking other causes of SCD, and it is less detailed in defining the exclusion criteria.

More recently, in 2020, Jessen et al. [6] proposed a framework for characterizing SCD to enhance the understanding and diagnosis of early AD. This proposal offers some advantages: (1) by identifying SCD as an early marker for AD, there is a greater opportunity for early intervention; (2) it emphasizes the importance of subjective experiences of cognitive decline as the earliest indication of neurodegenerative processes before objective measures detect abnormalities; (3) it provides standardized criteria for diagnosing SCD, which can enhance consistency in research and clinical practice; (4) it includes various factors such as age, education, and family history, offering a holistic view of the patient’s condition; (5) it encourages the integration of biomarkers; and (6) it contributes to the increasing public awareness about the early signs of cognitive decline, encouraging individuals to seek medical advice sooner. As in previous proposals, there are also problematic issues or risks. Some potential problems have been pointed out that would be related to overdiagnosis (many detections of false-positives, inconsistency due to subjective reports of cognitive decline, or cultural differences in these self-perceptions). Other criticisms involve the great number of resources needed for assessments that may not be feasible in all clinical settings. Additionally, informing patients about their SCD and its potential link to AD could increase anxiety and stress for them and their families.

While no single definition of SCD has achieved universal acceptance, further work towards a standardized definition will benefit from integrating these varied perspectives to improve the accuracy and utility of SCD as a diagnostic and research tool. In the present study, we used the Jessen et al. (2014) [4] criteria since the participants’ recruitment took place during 2019 and 2020, and provided that, on those dates, this proposal was more specific than any other.

Individuals with SCD should not exhibit clinically significant cognitive deficits on routine cognitive screening or testing, but there is increasing evidence showing subtle cognitive and neurobiological differences with matched healthy older adults that can be detected through specialized assessments [7]. Following this line of reasoning, a recent systematic review and meta-analysis conducted by Rotenberg et al. [8] showed a small but significant relationship between subjective and objective cognitive functioning in older adults. For their part, Jessen et al. [9] found that approximately 40% of individuals with SCD who consult memory clinics are in the Alzheimer’s disease (AD) continuum at stage 2, as indicated by positive amyloid biomarkers; consequently, this group showed extended subtle symptoms and an accelerated longitudinal cognitive decline. These results are in line with the longitudinal study carried out by Kryscio et al. [10], which identified that concretely subjective memory complaints were predictors of mild cognitive impairment within 9 years, although additional longitudinal studies are needed to better understand potential predictors of future objective cognitive declines [11]. Thus, the transition between DCS, MCI, and prodromal Alzheimer’s is a “gray zone” due to overlapping symptoms and variability in disease progression. This underlines the need for clear and consistent diagnostic criteria, as well as a multidimensional approach that considers both patient self-perception and objective assessments and biomarkers.

The evidence regarding those subtle differences between SCD and healthy older adults in different cognitive domains (executive functions, episodic memory, and language) is cumulative. For example, some studies have shown that older adults with SCD may exhibit subtle changes in working memory compared to their cognitively healthy counterparts [12]. Others have reported differences in episodic memory and executive functions between these two groups [13]. At this point, a relevant term from a neuropsychological perspective may be introduced. Discrepancy is a commonly used tool to show clinical differences between two values, sets of values, or conditions. Generally, a discrepancy involves subtracting one value from another [14]. In this context, “discrepancy” is a term related to the expectation about the result of a difference that is considered relevant from a neuropsychological point of view. However, the clinical and research utility of neuropsychological discrepancies has been explored in only a few studies. Throughout the manuscript, the term “discrepancy scores” (see for example: [15,16]) will be used to refer to differences that are of interest from a neuropsychological point of view and that can be calculated from different tests or tasks. There are some early precedents [17] that suggested, for example, that the difference between part A and part B might be more useful in interpreting the Trail-Making Test performance if the intent is to derive a relatively isolated measure of the shifting/executive functioning. A study conducted by Jacobson et al. [18] highlighted the importance of considering alternative methods of defining cognitive changes, such as evaluating cognitive discrepancy in people at risk for AD.

Another set of studies has suggested that the discrepancy between phonemic and semantic/categorical verbal fluency may be of diagnostic value [19] since the difference may reflect different sources of dysfunction. It is now well documented that even in the earliest stages, the semantic/categorical verbal fluency appears to be consistently impaired to a greater extent, suggesting the degradation of the semantic system [20]. Research in patients with MCI has described a similar pattern of verbal fluency impairment [21]. Considering normative data [22] healthy older adults demonstrated a significant advantage in the semantic condition relative to the phonemic condition. However, in the patient groups (MCI, AD), this advantage disappeared, with MCI patients showing only a non-significant semantic advantage, and dementia patients showing the reverse pattern, with a significant phonemic advantage. More recently, Chasles et al. [23] and Liampas et al. [24] have obtained similar results.

Regarding episodic memory, the Wechsler Memory Scale III (WMS-III; [25]) allows for the computation of 16 discrepancy comparisons using primary indexes. One of the most used is the auditory immediate–auditory delayed recall difference. The “reliable difference” approach is used for this type of test score evaluation [26]. Bruno et al. [27] reported the results of a study that investigated the effectiveness of several memory indexes derived from the Rey Auditory Verbal Learning Test in predicting the concentration level of biomarkers in an experimental sample that included MCI, SCD, and healthy controls. Consistent with previous findings reported by the same group, the recency ratio (a measure of the performance decline in recalling terminal items on the list passing from immediate to delayed trials) was the one most associated with the phosphorylated cerebrospinal fluid and total tau protein concentration.

Forward and backward digit span scores are commonly used neuropsychological tests of short-term verbal and working memory [28]. The forward digit span has been argued to represent a measure of the capacity of the phonological loop [29], whereas the successful performance on the backward version of the task represents a measure of central executive function due to the additional requirement of the manipulation of information within temporary storage [30]. There is a clinical perspective maintaining that with advancing age, the discrepancy between the forward and backward span increases, that is, the forward digit span tends to remain stable, whereas the backward digit span tends to decrease due to a decline in the central executive performance [31]. However, it is necessary to point out that there have also been inconsistent findings indicating that there is no greater age-related decline in the backward digit task than in the forward version [32]. Research suggests that SCD may be associated with changes in the discrepancy between forward- and backward-span tasks in comparison to healthy older adults. The evaluation of working memory in individuals with SCD using Forward Digit Span and Backward Digit Span tasks reveals potential alterations in cognitive functions compared to healthy older adults [33].

Regarding the language domain, and specifically in one of the tests most frequently used by clinicians, the Boston naming test (BNT), the total score results were obtained from adding the number of correct spontaneous responses plus the number of pictures named with the help of the semantic key. The correct answers after a phonological key are considered an indicator of the type of difficulty in naming [34,35]. If a person can successfully name an item with the help of phonological cues, it suggests that they have some problems accessing phonological forms spontaneously once the correct recognition and activation of the concept in the semantic system have been completed. Previous studies have shown a consistent relationship between the BNT total score and sociodemographic variables such as age and educational level. In this way, it has been observed that the naming performance decreases as age increases and years of schooling decrease [36,37,38]. A discrepancy score between the total score and the correct answers after a phonological key in BNT does make sense because it would be an index which is aware of difficulties in accessing phonological word representations [39]. Additionally, in this domain, there is also a type of task that is used less frequently within evaluation protocols, the sentence–picture verification paradigm, which aims to evaluate comprehension. This is a task that implies integrating a linguistic representation with an external representation of the event [40,41]. To serve the purpose for which it is designed, it is necessary to use only semantically reversible sentences where the use of syntactic cues is necessary for the correct representation of the argument structure. For example, in Spanish, the canonical (more frequent) word order is Subject + Verb + Object. Non-canonical structures (such as passives) require additional cognitive processing because constituents are moved away from their canonical (default) positions [42]. The difficulty associated with non-canonical reversible sentences is attributed to the complexity of assigning thematic roles to constituents displaced from their canonical position. This increased processing load can tax working memory and slow down comprehension. A discrepancy score (canonical word order–non-canonical word order) in a sentence–picture comprehension test would be informative given that the performance in non-canonical sentences is lower in patients with AD, SCD, or MCI, in comparison with healthy older adults [43,44,45].

The first main goal of this study is to explore the existence of significant differences in six discrepancy scores derived from three cognitive domains (memory, executive functions, and language) between a group of older adults with subjective cognitive decline (SCD+) and a group of older adults with normal cognition (SCD−). A second goal is to determine whether the selected discrepancy scores would have a significant role in the participants’ diagnostic classification. According to the literature review, the most common (although subtle) decline observed in older adults with SCD is mainly related to episodic memory and then executive functioning [46,47]. Thus, by hypothesis, it is expected that the most significant differences between groups in the discrepancy scores will be related to these domains. Following the same line of reasoning, it is expected that dependent measures related to episodic memory and executive functions would have a more relevant role in the subjects’ classification than the other dependent measures.

## 2. Materials and Methods

### 2.1. Participants

A total of 75 Spanish-speaking older adults voluntarily participated in the present study. An incidental sampling (of convenience) was used. The researchers contacted the directors of residences and health centers in the city of Madrid (Spain) to authorize us to put up posters requesting volunteers for a neuropsychological study on aging and to inform people they considered who could potentially meet the inclusion criteria. Interested people wrote to the researchers using an email address or via WhatsApp, which allowed us to make the first appointment to carry out the initial screening. All participants gave their written informed consent to participate in the study, which was approved by the Hospital Universitario San Carlos de Madrid local Ethics Committee (internal code: 18/422-E_BS) and was completed following the Helsinki Declaration. No one had a history of neurological or psychiatric disorders. From the total sample, 32 participants formed the SCD+ group and 43 belonged to the control group (older adults with normal cognition: SCD−). All participants had normal or corrected hearing and vision. Table 1 shows the demographic and clinical characteristics of the participants. The groups did not differ in any of the summarized variables.

### 2.2. Design and Materials

A complete neuropsychological assessment provided the relevant information for the classification of the total sample; in this sense, the following instruments were applied for diagnostic purposes. A questionnaire to collect socio-demographic and health information was first applied. The assessment protocol included the Mini-Mental State Exam (MMSE; [48]) as a brief cognitive screening test, the delayed paragraph recall in the Logical Memory subscale from the Wechsler Memory Scale III (WMS-III; [25]), and the Geriatric Depression Scale (GDS-15; [49]). The administration of the GDS-15 was important, given that SCD can be associated with depressive symptoms [50].

All participants, both normal cognition (SCD−) and SCD+ older adults, met the following criteria according to normative Spanish data: (1) Mini-Mental State Examination (MMSE) ≥ 27 points indicating normal cognitive functioning, (2) GDS-15 score ≤ 6 points, and (3) normal scores in WMS-III’s delayed paragraph recall (units) according to age (55–65 years ≥ 18; 66–73 ≥ 12; 74 or more ≥ 9). Most of the participants with normal cognition (SCD−) were relatives of SCD+ participants (husbands, wives, brothers, and sisters) who decided to accompany them on the same days they went to the evaluation appointments. The remaining participants in the control group were neighbors or coworkers of the individuals with SCD+.

Additionally, SCD+ participants, according to the criteria proposed by Jessen et al. [4], were those who had a self-perception of progressive deterioration in cognitive functioning in the absence of objective evidence of cognitive decline and depressive symptoms, as stated above. Thus, these participants (a) had a requested medical consultation because of their memory complaints, (b) felt that their subjective decline affected their daily activities, (c) set the onset of their subjective decline within the last 2 years, and (d) had concerns associated with their subjective decline which were confirmed by a reliable informant. None of these patients met the criteria for MCI and had no history of psychiatric or neurological disorders according to clinical records.

Other tests were used as key dependent measures to compute discrepancy scores for the analysis, as stated in the Introduction. In this sense, the assessment protocol included letter (F, A, S) and categorical (animals, fruits) verbal fluency tasks [51], the WMS-III Word List (immediate and delayed recall of units), and Digit Span (forward and backward) subtests [25], the 60-item Boston Naming Test (BNT; Adapted Spanish edition by García-Albea et al. [52]), the Stroop test [53], and the ECCO test (Exploración Cognitiva de la Comprensión de Oraciones; English Translation: Cognitive Assessment of Sentence Comprehension [41]). This last test allows us to assess the thematic role assignment (“who did what to whom”) with a set of 36 sentence–picture pairs including a variety of sentence structures fitted or not to canonical word order in Spanish (a Subject + Verb + Object [SVO] language).

### 2.3. Procedure

All tests mentioned above were part of a larger evaluation protocol that was used in a research project on typical and pathological aging. In the first session, the sociodemographic questionnaire was completed, and the screening tests were also applied. The rest of the tests (memory, language, and executive skills) were applied in the second session along with other neuropsychological tests and questionnaires selected for the broader study. The interval between the two sessions ranged from 5 to 10 days. All the neuropsychological tests were administered following the instructions in their respective manuals.

### 2.4. Statistical Analysis

As a previous step to the analyses, a set of discrepancy scores were computed. Regarding verbal fluency (VF), first, the mean scores were computed for the letter or phonemic (F, A, S) and semantic (animals, fruits) tasks, and then the difference of the semantic–phonemic VF. Regarding Stroop, the discrepancy score was obtained by simply subtracting the individual’s score on the interference condition from the score of the Word condition (Word condition-interference condition). In the case of the WMS-III Word List, the difference was used: immediate–delayed recall. The discrepancy score for the WMS-III Digit Span was obtained by the following difference: digits forward–backward. BNT was computed with the calculation of the number of correct spontaneous answers—the number of correct answers following a phonemic key. Remember that a “spontaneous response” refers to the immediate and unprompted verbal identification of an item presented in a picture by the test-taker; a “correct answer following a phonemic key” occurs when the test-taker correctly identifies the object after being given the initial sound or sounds of the word (used to assist individuals who may have difficulty retrieving the correct word spontaneously). Finally, a discrepancy score related to sentence comprehension (ECCO test) was obtained considering the following difference between the total number of correct responses in two sets of sentences: fitted in canonical word order in Spanish (WOS)—not fitted in canonical WOS. In the simple verification task used in the ECCO test, the participants respond correctly by saying YES when the image and the phrase matched perfectly, or by saying NO, if the image and the phrase did not match.

Descriptive statistics were included for all dependent measures. A multivariate ANOVA was used to find significant differences between the groups and eta squared partial as an estimation of the effect size. The difference between the groups on a discrepancy score was considered statistically significant when the *p*-value was less than 0.05. The binary logistic regression was computed using the discrepancy scores that were significant in the previous ANOVA as predictors to establish the participants’ classification. To assess the role of each predictor in the logistic regression, we considered the regression coefficients (B), the *p*-values, 95% confidence intervals, and odds ratios (ExpB). When *p* < 0.05 and the confidence interval does not include the value 0, the predictor variable is considered significant. A positive β coefficient indicates an increase in the probability of the outcome, while a negative coefficient indicates a decrease. An ExpB greater than 1 indicates that as the predictor variable increases, the probability of the outcome also increases, and an ExpB less than 1 indicates the opposite.

A final ROC curve analysis was also performed with the variables that had an important role in the subjects’ classification to obtain cut-off values with potential clinical significance. An ROC curve is a graphical representation of the trade-off between sensitivity and specificity for a binary classification test at various threshold settings. The curve is plotted with Sensitivity (True Positive Rate) on the *y*-axis, and 1-Specificity (False Positive Rate) on the *x*-axis. Each point on the ROC curve corresponds to a different threshold used to classify a positive case. Although the SPSS automatically determines the cut-off values corresponding to an ROC curve, in summary, this procedure implies (1) the calculated cumulative distributions of the true positive rate (TPR or sensitivity) and the false positive rate (FPR or 1-specificity) in each threshold. (2) The identification of the threshold that corresponds to the maximum K-S statistic (which is the absolute difference between the TPR and FPR). This threshold represents the point where the difference between the cumulative distributions of the true-positives and false-positives is greatest. (3) The threshold at which the maximum K-S statistic occurs is considered the optimal cut-off value for classification since this value maximizes the separation between the positive and negative classes.

## 3. Results

The study’s results are described in the following three subsections: the discrepancy scores and differences between groups, participants’ classification with significant variables, and ROC curves.

### 3.1. Discrepancy Scores and Differences between Groups

A summary of the descriptive statistics for dependent variables as well as the result of a multivariate ANOVA are shown in Table 2.

Significant differences between the groups are observed in the naming discrepancy score (BNT spontaneous responses—phonetic key evoked responses), as well as in the sentence comprehension discrepancy score (ECCO sentences fitted in canonical WOS—not fitted in canonical WOS). The effect sizes in these two dependent measures could be considered large (greater or equal to 0.014). The remaining differences did not reach statistical significance (*p* ≥ 0.321); the effect sizes for all these dependent measures were small (lower or equal to 0.01).

### 3.2. Participants’ Classification

A binary logistic regression was computed using both the BNT and the ECCO discrepancy scores, associated with the language domain. This analysis reached 71.6% correct classification, as can be seen in Table 3.

The final equation included the two discrepancy scores, as shown in Table 4. The negative beta corresponding to the BNT could be interpreted as a near zero correlation with subjects’ classification, while in the case of the ECCO, it would be medium/moderate, suggesting that the higher discrepancies in the test are associated with better subject classification. 

Exp(B) represents the odds ratio associated with a one-unit change in the predictor variable. When Exp(B) is less than 1, as in the BNT discrepancy score, it indicates that the increasing values of the predictor correspond to the decreasing odds of the event’s occurrence. However, regarding the ECCO discrepancy score, Exp(B) suggests that for every one-unit increase in the predictor variable, the odds of the event happening (correct classification) are approximately 1.5 times higher. This implies a positive relationship between the predictor variable and the likelihood of the event.

### 3.3. ROC Curves

Although no differences were found between the groups regarding the discrepancy scores derived from the following neuropsychological tests, nor did they contribute significantly to the classification of the participants, these are their diagnostic accuracy results (area under the curve: AUC): AUC_Verbalfluency_ = 0.572; AUC_WordList_recall_ = 0.505; AUC_DigitSpan_ = 0.590; and AUC_Stroop_ = 0.505.

ROC analysis using the BNT discrepancy score and SCD+ as the positive success criteria showed an area under the curve (AUC) of 0.690, 95% CI [0.564; 0.815], *p* = 0.003. This value means that the BNT discrepancy score has a poor-to-fair diagnostic accuracy. Figure 1 illustrates the curve related to the diagnostic accuracy.

The Kolmogorov–Smirnov (K-S) metric offers an index showing how far separated the rate of true positives is from the rate of false positives, thus indicating if the model is good enough for classification. In this case, K-S was equal to 0.376, indicating that the model has a medium/moderate quality to distinguish between the two groups. The optimal cut-off for the BNT discrepancy score based on the K-S metric for our sample was 48.5. Concerning the ECCO discrepancy score, ROC analysis using SCD− as the positive success criteria showed an area under the curve (AUC) of 0.722, 95% CI [0.603; 0.840], *p* < 0.001. In this case, the AUC reflects that the ECCO discrepancy score has a fair diagnostic accuracy, that is, the classifier performs adequately but may not be suitable for all diagnostic purposes; it has a reasonable discriminative ability but might require improvement. Figure 2 illustrates the ROC curve relating sensitivity and the 1-specificity.

K-S was equal to 0.366, indicating again that the model has a medium/moderate quality to distinguish between the two groups. The optimal cut-off for the ECCO discrepancy score based on the K-S metric for our sample was 2.5.

## 4. Discussion

The main goal of this research was to identify significant differences between SCD+ and SCD− in six discrepancy scores derived from three cognitive domains (memory, executive functions, and language) and then, to determine what discrepancy score or scores would have a significant role in participants’ diagnostic classification.

Regarding the first hypothesis, the results of this study showed significant differences between SCD+ and SCD− in the discrepancy scores related to language, specifically in naming (BNT) and sentence comprehension (ECCO). Thus, the results contradict what was expected. It must be considered that the groups did not differ in age, years of education, episodic memory, global cognitive state, or mood.

Concerning the second hypothesis, the two discrepancies mentioned above allowed a reliable participants’ classification (71.6%), with the ECCO discrepancy score having a medium/moderate positive relationship with the likelihood of the classification, contrary to what was expected given the second hypothesis. The odds ratio indicates for this discrepancy score that for every one-unit increase in the predictor, the correct classification is approximately 1.5 times higher. The ROC curve analyses pointed out that individual models using the two discrepancy scores mentioned above are poor-to-fair for classification, as shown by the values of the area under the curve (AUC) and the Kolmogorov–Smirnov (K-S) metric. The K-S metric provides cut-off points for both discrepancy scores, the one derived from BNT (=48.5) and the other corresponding to the ECCO test (=2.5), respectively. The BNT’s discrepancy score will be usually positive, optimally equal to 60 points, and thus a progressive decrease could be interpreted as a progressive increase in the subject’s dependency on a phonological key to access the word form for correct naming. Regarding ECCO’s discrepancy score, it will be usually positive, optimally equal to zero (since there are 18 canonical and 18 non-canonical structures in the test), and thus a progressive increase could be interpreted as indicating that the subject’s comprehension of non-canonical structures is progressively decreasing.

In the scientific literature, results about the differences between SCD+ and SCD− maintain inconsistency. Evidence for subtle but significant neuropsychological differences between SCD+ and SCD− older adults are far from conclusive since some cross-sectional performances or longitudinal follow-up studies have reported no differences between these groups in a range of cognitive functions including memory, language, executive functioning, and global cognition [54,55,56], but others have observed that SCD is associated with a lower cognitive performance in the same cognitive domains [57,58,59]. In addition to the studies already mentioned, the current work provides more evidence showing slight but significant differences between SCD and control older adults.

López-Higes et al. [42] reported results from a study comparing Spanish SCD and healthy older adults and exploring the role of demographic and executive factors in predicting the linguistic performance in these groups. The analysis revealed that the control group’s average scores were significantly higher than those obtained by the SCD group in BNT naming, like in the present study, but both groups reached similar results in the four derived measures related to the ECCO sentence comprehension test (different from the discrepancy score used here). However, the performance in the most complex items (non-canonical reversible sentences with two propositions) was mostly explained by flexibility in the control group and by the inhibition efficiency in the SCD group.

For example, Wolfsgruber et al. [7] analyzed the baseline sample of the DELCODE study and reported that, although SCD older adults presented an unimpaired overall cognitive performance, they exhibited minor deficits in memory, executive function, and language abilities. The authors found an association of these subtle cognitive deficits with Alzheimer’s Disease cerebrospinal fluid biomarkers and concluded that these subtle differences may have validity and potential use for the early detection of underlying preclinical AD. Regarding language, the authors found a subtle but significant deviation of participants with SCD from the healthy older adults in a composite factor including the BNT-15 supplemented by five infrequent items from the long version, the naming part included in the Free and Cued Selective Reminding Test, and two fluency tasks (animals and groceries), a result that is compatible with what was found in the present study.

More recently, Jessen et al. [9] reported the results of cross-sectional and longitudinal data from the DELCODE study, which they conducted with people aged 60 years or older. To derive factor scores of learning and memory, language ability, executive functions and processing speed, working memory, and visuo-spatial abilities from the neuropsychological test battery in DELCODE, they applied the same confirmatory factor analysis method described in Wolfsgruber et al. [7]. The SCD group showed slightly worse cognition as well as subtler functional and behavioral symptoms than the control group. At the baseline, among other results that indicate significant differences between the groups, the SCD group showed a worse performance than the control group’s factor scores on memory, language abilities, and executive function.

Morrison and Oliver [59] followed up annually 3019 normal older adults for a maximum of 15 years, including 831 participants with SCD. Their results pointed out that people with SCD exhibited lower baseline scores and a steeper decline in global cognition, episodic memory, semantic memory, and perceptual speed. This group did not differ from SCD− in the baseline visuospatial ability or working memory, but exhibited an increased change over time in those two domains in contrast to SCD−. The authors suggested that older adults with SCD may be aware of subtle cognitive declines that occur over time in all those previously mentioned domains.

A study conducted by Malyutina et al. [60], with 163 participants from 55-to-93-year-old memory clinic patients, made a novel contribution to the emerging research on these topics by using measures of language performance (naming-by-definition and sentence comprehension) and a detailed self-assessment of subjective language complaints (SLCs) that assessed not only word retrieval, but also language comprehension during reading and listening. The results showed that greater SLCs were significantly associated with slower naming latency from the naming-by-definition task and lower sentence comprehension accuracy from the sentence comprehension task. Therefore, apart from the differences relative to the sample used in their study, the results presented are quite similar to those of our study.

As a final remark, our results pointed out that language objective discrepancy scores, especially ones related to sentence comprehension, are important to distinguish between older adults with SCD and those with normal cognition, but also highlight the clinical importance of using a detailed self-assessment addressing specific components of the language function to correctly characterize cognitive complaints. As revealed in the literature, this research field is complex and has revealed inconsistent results. Given the importance of detecting early signs that cognitive ability is deteriorating in older adults, more studies are needed, particularly in aging populations, as in the case of Spain.

There are limitations of the current study that should be noted. The main threats of bias in the selection of the study’s incidental sample would be the lack of representativeness and possible self-selection (probably only those who have the time and desire to participate responded). For example, the education levels of the participants are medium/high and thus may not be generalizable and may have contributed to not being able to confirm some of the hypotheses. Additionally, although groups did not differ in relevant demographic and clinical variables, the groups’ sample sizes were unbalanced (32 vs. 43 participants). A related issue is that a sample size calculation was not performed, which is another limitation. Furthermore, the sample should be more diverse. It would even be interesting in future studies to have a more representative sample from Spain and then from other countries. It would also be interesting for future studies to make a comparison between older adults with and without complaints and patients with mild cognitive impairment. It would be important to also compare middle-age groups with older age groups. Also, as mentioned in several studies [11,61], it would be important to conduct not only transversal, but also longitudinal studies, combining behavioral, neuroimaging, and self-report data.

The study explored which cognitive tests and domains help differentiate those who are labeled as SCD+ and SCD−, which may have implications for future research examining how to identify those more likely to progress to MCI [62]. However, further studies should be carried out to reinforce the results that have been reported.

## 5. Future Directions

At this point, it is important to reflect on certain aspects related to the method used in the study to determine the cut-off points concerning the discrepancy scores that are of greatest interest and to consider how their clinical relevance could be evaluated in future work. Although these methods are crucial for understanding and optimizing the performance of a diagnostic or classification test [63], the clinical assessment of the cut-off values would involve following at least four steps in future studies: (1) Analyze the sensitivity, specificity, positive predictive value [64] (it takes into account both the sensitivity and specificity of the test, as well as the prevalence of one condition of interest in the population being tested), and negative predictive value at the K-S cut-off. (2) Conduct studies or simulations to assess how the cutoff performs in clinical practice. (3) Engage with clinicians and stakeholders to review findings and gather feedback. Finally, (4) implement the cut-off value in clinical practice and continuously monitor its impact on patient outcomes and clinical workflows.

A possible future direction that could be derived from our study would involve the development of normative data on the discrepancy scores that we have used for older people between the ages of 60 and 80, in line with some current works [16]. These normalized discrepancy scores could be used not only to compare an individual with his or her group considering sociodemographic variables, but also to determine the deviation that a person with a given diagnosis presents concerning the group that presents typical aging, or the change that occurs in the discrepancy (between tests or intra-test) after an innovative proposal which, for example, allows a personalized intervention to the state of each individual (e.g., cognitive state) in the geriatric population [65].

There are actual evidence-based diagnostic methods that have clinical and research applications in neuropsychology and constitute a potential line of development in future studies. For example, Bayesian models facilitate clinical decision making in different ways [66]: (1) computing the post-test probability that a test score belongs to a set of diagnostic possibilities; (2) estimating how a test score profile compares scores across various diagnoses; and (3) obtaining the probability of achieving individual scores, discrepancies (between tests or within tests), or other performance-based summaries within the population of interest.

## Figures and Tables

**Figure 1 geriatrics-09-00083-f001:**
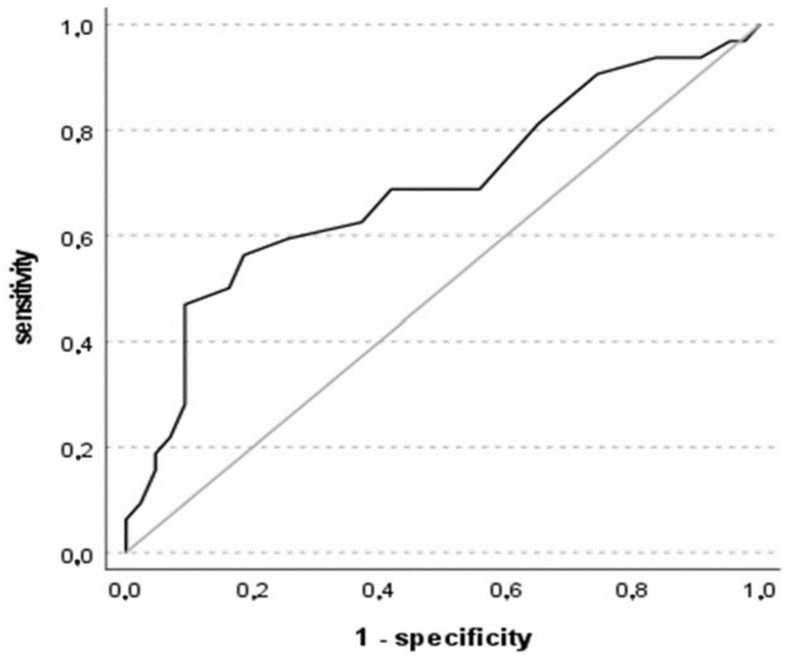
ROC curve for the BNT discrepancy score. The reference line is in grey.

**Figure 2 geriatrics-09-00083-f002:**
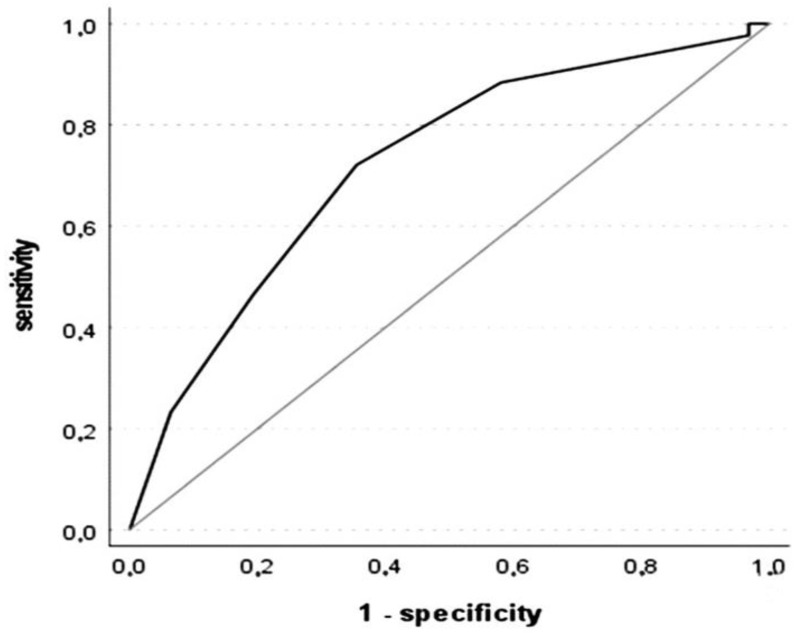
ROC curve for the ECCO discrepancy score. The reference line is in grey.

**Table 1 geriatrics-09-00083-t001:** Participants’ demographic and clinical characteristics. SCD+: Older adults with subjective cognitive decline.

	Normal Cognition (SCD−)		SCD+		*F* (1, 69) =	Sig.
	Mean (SD)	Range	Mean (SD)	Range		
Age	69.81 (4.62)	60–78	71.50 (5.29)	62–80	2.01	0.161
Years of education	14.07 (5.44)	6–20	12.54 (5.45)	4–20	1.35	0.250
WMS-III Logical Memory units delayed recall	24.91 (8.09)	12–39	21.71 (8.82)	10–40	2.46	0.122
MMSE	28.95 (1.11)	27–30	28.62 (0.94)	27–30	1.16	0.285
GDS-15	1.02 (1.41)	0–5	1.36 (1.47)	0–6	0.922	0.340

**Table 2 geriatrics-09-00083-t002:** Descriptive statistics (mean and standard deviation) of dependent variables across the groups and multivariate ANOVA results.

Discrepancy Scores	SCD−	SCD+	*F* (1, 72) =	Sig.	Effect Size
	Mean (SD)	Mean (SD)			
Verbal Fluency	2.79 (4.01)	3.69 (3.56)	1.00	0.321	0.014
Word List_recall	23.02 (4.58)	23.13 (4.78)	0.01	0.924	0.000
Digit Span	2.63 (1.84)	3.06 (1.89)	0.989	0.323	0.014
BNT	51.74 (5.42)	46.64 (8.11)	10.51 **	0.002	0.127
Stroop	66.65 (15.26)	67.80 (13.74)	0.11	0.739	0.002
ECCO	1.79 (1.52)	3.09 (1.71)	12.17 **	0.000	0.145

** (*p* < 0.05).

**Table 3 geriatrics-09-00083-t003:** Results of participants’ classification.

		Predicted		Correct Percentage
		SCD−	SCD+	
Observed	SCD−	36	7	83.7
	SCD+	14	17	54.8
Global percentage				71.6

**Table 4 geriatrics-09-00083-t004:** Summary table showing the results obtained in logistic regression.

Discrepancy Scores	B	Standard Error	Wald	gl	Sig.	Exp(B)	95% C.I. for EXP(B)	
							Inferior	Superior
BNT	−0.091	0.04	4.76	1	0.029	0.91	0.84	0.99
ECCO	0.415	0.17	5.99	1	0.014	1.51	1.08	2.11
Constant	3.18	2.18	2.12	1	0.14	24.02		

## Data Availability

The data that support the findings of this study are available from the corresponding author (P.F.S.R.).

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
