# Peer review of "Differentiation between Normal Cognition and Subjective Cognitive Decline in Older Adults Using Discrepancy Scores Derived from Neuropsychological Tests"

_geriatrics, 2024, doi:10.3390/geriatrics9030083_

Round 1
Reviewer 1 Report
Comments and Suggestions for Authors
The submitted article deals with an interesting theme, focusing on discrepancies in naming and sentence comprehension between healthy and older adults. The conceptual framework of the article is logical, clearly demonstrating that the authors have chosen a topic with significant practical relevance and have focused on specific themes. This is evidenced by the use of a range of professional sources, overall making the article appear to be of high quality. To further improve, I suggest the following points:
I could not find an answer in the text to: The study reportedly examined differences in six discrepancy indexes derived from three cognitive domains (memory, executive functions, and language), but the abstract lacks a detailed explanation of the specific aspects of these domains that were studied. Adding a brief description of each index could help readers better understand the scope and focus of the study.
While the abstract provides information that significant differences were found and certain indexes allow for reliable classification of participants, it would be useful to place these results within the broader context of current SCD research. If possible, a brief mention of how your findings differ from or complement previous work in this area could add value.
A more precise description of the study participants (for example, "older adults" is mentioned, but more specific information about demographic characteristics would be useful for readers) and methods could enhance the study's transparency and replicability.
When stating that the ROC curves show "medium/moderate quality" for distinguishing between groups, it would be beneficial for readers if you provided specific AUC (Area Under the Curve) values or other metrics that support this quality.
It would also be useful to mention the potential practical applications of your findings in the abstract. How can your results aid in clinical practice or in developing interventions for individuals with SCD? If you are limited by the abstract, I would emphasize such recommendations in your conclusions + justify the study's limitations (I recommend mentioning comprehensive approaches for geriatric clients at the end of your paper. For example, I recommend using the general principles from the attached articles - https://www.webofscience.com/wos/woscc/full-record/WOS:000528487500007 and https://www.webofscience.com/wos/woscc/full-record/WOS:000801505500001)
Clarity of index calculation methods: The description of how discrepancy indexes were calculated for various cognitive tests is well-structured and provides clear information on the methodology. However, for some indexes (e.g., BNT and ECCO test), it would be helpful to add more details about what exactly is meant by "correct spontaneous answers" or "correct answers following a phonemic cue" to better understand the methodology.
Statistical methods: The use of descriptive statistics, MANOVA, binary logistic regression, and ROC curve analysis is appropriate for the types of analyses conducted. However, it would be useful if the criteria for selecting variables for the binary logistic regression based on MANOVA results were described in more detail, including how significant discrepancy indexes were determined and how their role in participants' classification was assessed.
Interpretation of effect size: Mentioning eta squared partial as an estimate of effect size is beneficial for understanding the significance of the differences found. However, adding an interpretation of these values would help readers better grasp the practical significance of the findings. For example, what values of eta squared partial are considered small, medium, or large effects in the context of this study?
ROC curves and clinical significance: Mentioning ROC curves and determining cutoff values with clinical significance is important for applying the results in practice. However, it would be advantageous to discuss more thoroughly how these cutoff values were determined and how their clinical relevance was evaluated, including a discussion on sensitivity, specificity, and possibly other relevant metrics, such as accuracy or positive predictive value.
I ask the authors to consider revising these points.
Primarily, I would focus on processing and formulating better conclusions and recommendations for practice. An integral part of such recommendations are the study's limitations, which need to be addressed and highlighted. If the authors reflect on these recommendations, I recommend the article for publication. I believe in the broad impact of this article.
Author Response
Dear Reviewer,
Please, find in the attached file our responses.

Reviewer 2 Report
Comments and Suggestions for Authors
Geriatrics 2961468
Thank you for the opportunity to review this interesting manuscript. I have a number of comments/questions:
Abstract/Title:
The title is ill-defined and not precise. It should be completely re-written. I presume the authors mean differentiate when they say “make the difference”. Also at the end of the title, please rephrase the title to read “between older adults with normal cognition and subjective cognitive decline”. Healthy could refer to any/all aspects of health when I presume they are only referring to cognition.
There are a number of grammatical errors in the abstract and throughout the manuscript which detract from the reading of the paper – an English language edit is strongly advised. The abstract in particular is difficult to read and I had to do so multiple times to really understand what the summary findings of the paper are.
Throughout the abstract and main manuscript avoid the term “SCD individuals” (line 14) and say “individuals with SCD”.
Line 14: Typo: Individuals present “with” differences (line 14).
Explain “discrepancy index” in the abstract – readers will not likely be familiar with this term. How did the authors come up with 6 of them?
Line 20: Specify that the control group is the SCD- group.
The term “medium/moderate quality” is not scientific. What do they mean by quality to distinguish? Do they mean diagnostic accuracy? ROC curve analysis provides AUC scores denoting diagnostic accuracy, which can be interpreted as fair through to excellent. These should be provided/summarised in the abstract to show that naming and sentence comprehension help differentiate those with and without SCD and how well they do this.
Based on the AUC values presented in the main manuscript in section 3.3., the diagnostic accuracy of the individual cognitive tests used to differentiate SCD+ from SCD- (e.g. the Boston Naming Test) are poor to fair at best (0.69 to 0.72).
Main manuscript:
Please describe/define SCD from the outset. There is no accepted standardised gold standard definition so it is important to understand what the authors are referring to. I see below that the authors use Jessen et al to define SCD in this analysis but they should mention several of the definitions and the pros and cons in the introduction.
As the authors point out – “SCD may not exhibit clinically significant cognitive deficits”. Please specify that this is on routine cognitive screening or testing. I would also say “should” rather than “may” as by any standard definition, it should not.
If cognitive deficits are identified in the presence of subjective cognitive symptoms, is this not essentially a diagnosis of MCI provided dementia/functional impairment are not present? If they are biomarker positive for Alzheimer’s disease (AD), then they have prodromal AD. In this sense SCD could be reserved for those with symptoms but who have normal neuropsychological testing and who are biomarker negative. It is a grey area.
Line 56 define a “commonly used form”.
Line 57 define “discrepancy scores”. Do the authors mean cognitive test or subtest scores or individual cognitive domains such as episodic memory or language etc.? If this is the case, please stop using the term discrepancy score here and throughout the manuscript. It is not a commonly used term in English.
Line 67 – again “discrepancies” is used – in English, a discrepancy is “an illogical or surprising lack of compatibility or similarity between two or more facts.” Do the authors simply mean that a “difference” in phonemic and semantic/categorical verbal fluency may be of diagnostic value???
How were patients sampled? What method was used/ I presume this is a form of convenience sampling? Did they self-select? Who recruited them? Could this small sample be prone to selection bias?
Describe who the SCD- participants are – I presume these did not attend clinic as they did not meet item (b) seek out medical attention. Who are these controls? How were they recruited?
Line 176 – where is (a)?
Line 258 – indicate that an AUC curve value of 0.69 is poor to fair at differentiating the SCD+ and SCD- groups.
Line 259 – Please replace “sensitivity and the 1-specificity” with diagnostic accuracy – the latter is correct term.
Please provide the AUC values for verbal fluency, the stroop test etc. - they may turn out to have similar or better AUC values as the BNT and ECCO, even if there were no statistically significant differences in their mean values between the two groups.
Line 291 – Rather than “good enough” please write poor to fair. That is how to interpret ROC curve analysis and AUC values.
Another limitation, likely relates to the sampling approach – see my comment/query above.
I am assuming that a sample size calculation was not performed, which would be another limitation.
Line 382 – I would not say that this study has increased “knowledge about the association between subjective cognitive decline and the risk of clinical progression [58] to mild cognitive impairment in cognitively healthy individuals.” – the study did not show anything about the risk of progression to MCI – please state instead that the study explored which cognitive tests and domains help differentiate those who are labelled as SCD+ and SCD-, which may have implications for future research, examining how to identify those more likely to progression to MCI.
Comments on the Quality of English LanguageThere are a number of grammatical errors in the abstract and throughout the manuscript which detract from the reading of the paper – an English language edit is strongly advised. This is especially true for the abstract.
Author Response
Dear Reviewer,
Please, find in the attached file, our responses.
Round 2
Reviewer 2 Report
Comments and Suggestions for Authors
Thank you for revising the manuscript in detail.
A few minor comments:
In the abstract specify that it is "subtle differences in COGNITION BETWEEN individuals with subjective cognitive decline (SCD) compared to those with normal cognition."
Line 18-19 of the abstract: change "participated, with in the SCD+ group" to "were included: 32 who were labelled SCD+".
At the end of the abstract add that while these discrepancy scores are important, given their relatively poor diagnostic accuracy, that they should be used with caution as part of more detailed neuro-psychological assessment.
The introduction contains much more detail and reads better. Thank you.
Thank you also for better defining what a discrepancy score is.
The methods including how the sample was obtained and who the controls were is now clear. Thank you.
Comments on the Quality of English LanguageThe English language edit has improved the grammar and the paper reads better.
Author Response
Dear Reviewer/Dear Editor,
We appreciate the positive assessment. You can find in the attached file, our responses to the last comments.
Best regards.
